

# Transport through interacting defects and lack of thermalisation

**Giuseppe Del Vecchio Del Vecchio**[1⋆], **Andrea De Luca**[2] and **Alvise Bastianello**[3,4]

**1** Department of Mathematics, King's College London, Strand WC2R 2LS
**2** Laboratoire de Physique Théorique et Modélisation (UMR 8089),
CY Cergy Paris Université, CNRS, F-95302 Cergy-Pontoise, France
**3** Department of Physics and Institute for Advanced Study,
Technical University of Munich, 85748 Garching, Germany
**4** Munich Center for Quantum Science and Technology (MCQST),
Schellingstr. 4, D-80799 München, Germany

⋆ giuseppe.del_vecchio_del_vecchio@kcl.ac.uk

## Abstract

We consider 1D integrable systems supporting ballistic propagation of excitations, perturbed by a localised defect that breaks most conservation laws and induces chaotic dynamics. Focusing on classical systems, we study an out-of-equilibrium protocol engineered activating the defect in an initially homogeneous and far from the equilibrium state. We find that large enough defects induce full thermalisation at their center, but nonetheless the outgoing flow of carriers emerging from the defect is non-thermal due to a generalization of the celebrated Boundary Thermal Resistance effect, occurring at the edges of the chaotic region. Our results are obtained combining ab-initio numerical simulations for relatively small-sized defects, with the solution of the Boltzmann equation, which becomes exact in the scaling limit of large, but weak defects.

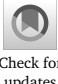
# 1 Introduction

The problem of thermalisation is a paramount question for many facets of physics at the center of hectic research. The advent of exquisitely precise experimental techniques [1–4] in engineering and probing non-equilibrium quantum states of matter spurred a deep interest in an apparently simple question: under which conditions a closed system thermalises?

It was believed for long-time that physical systems involving a large number of components are generally doomed to thermal equilibrium. Nowadays, mechanisms to escape thermalisation have been identified, such as many-body localization [5], integrability [6], scarring [7,8] and fragmentation [9]: in all these cases, the system does not fulfill the standard Eigenstate thermalisation Hypothesis (ETH) [10]. However, beyond spectral properties which are directly connected to homogeneous settings, it is natural to investigate thermalisation problems in inhomogeneous scenarios. An ideal laboratory to investigate transport phenomena is offered by the one dimensional world: here, efficient numerical algorithms [11] and powerful analytical methods have unveiled a plethora of exciting phenomena, as the suppression of transport in disordered [12] and confined [13–15] systems, ballistic transport in exactly solvable models [16, 17] with the possibility of diffusion [18, 19] and superdiffusion [20–22]. In the 1d world, the role of defects (or impurities) is enhanced since carriers move on a line and cannot avoid scattering. This setup attracted a great deal of attention, starting from the celebrated Kane-Fisher problem [23, 24] and with a renewed interest in recent times, in particular in the context of mobile impurities [25–33].

Another relevant aspect of impurities concerns their scrambling properties when embedded in an otherwise non-thermalising environment [34–38]. For the sake of concreteness, let us consider a bulk Hamiltonian $\hat{H}_{\text{bulk}}$ chosen to be integrable, thus supporting ballistic transport and prepare the system in an homogeneous non-thermal stationary state of $\hat{H}_{\text{bulk}}$. At $t \geq 0$, the Hamiltonian is locally perturbed and the system evolves in the presence of a defect $\hat{H} = \hat{H}_{\text{bulk}} + \hat{V}$, with $\hat{V}$ being constant in time and supported on a finite region around $x = 0$. Previous studies focused on the case where the defect preserves some notion of integrability [39–43], or the volume is finite [44–49]. In the latter case, the boundaries scatter the carriers back to the defect, eventually leading to thermalisation. This is seen also in the spectral properties of the finite dimensional matrix $\hat{H}$, where chaotic behavior immediately emerges for arbitrarily weak $\hat{V}$ [50–53]. On the other hand, integrability-breaking defects in extended systems remain widely unexplored.

As a consequence of the activation of $\hat{V}$, a perturbation spreads ballistically inside a light-cone centered around the defect [39]. The phase-space distribution of the carriers flowing out of the defect is of central interest: these excitations are *scrambled* when passing in the defect region and it might seem natural to assume a thermal distribution. While this has been contradicted for small impurities [35], the scenario is less clear at the mesoscopic scale, lying between impurity physics and thermodynamics: a strongly-interacting extended defect is itself a macroscopic system obeying the laws of thermodynamics. Hence, if a stationary state is reached, thermally distributed carriers should be emitted in the system's bulk.

Despite the extensive research in quantum systems, the key-point behind this question is the competing effects between integrability and its breaking within a finite, but extended, region. Importantly, the analogue scenario can also be engineered within a classical framework. Reverting to classical physics has the main advantage of being able to efficiently simulate system sizes and time scales that are far beyond the capability of present quantum numerical methods [11], especially in the case of highly excited and interacting states. Indeed, classical models recently gave important insights and benchmarks in several transport scenarios connected with integrability and originated first within the quantum context [54–57] and in this work we will walk through this path as well. Hence, we look at classical systems as an

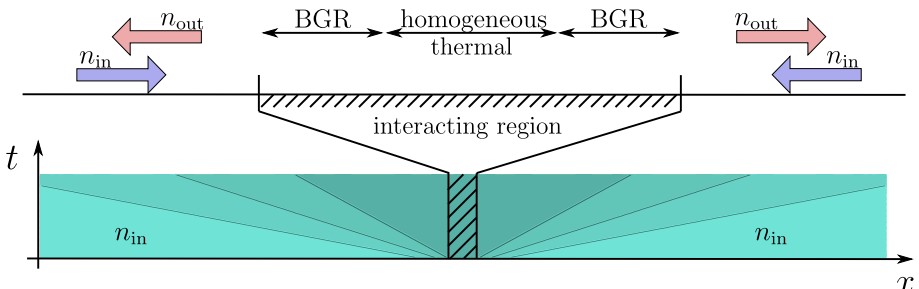

Figure 1: A nonequilibrium distribution of carriers $n_{\text{in}}(k)$ is injected on the defect, which at late times acts as a source of carriers spreading ballistically in the bulk (bottom). The outgoing carrier's distribution $n_{\text{out}}(k)$ is non-thermally distributed due to the presence of non-thermal regions at the edges of the interacting defect (BGR).

irreplaceable laboratory to guide our physical intuition, but we frame the very microscopic mechanism behind our observations within a kinetic picture whose validity is expected to extend to quantum systems as well.

In this work, we focus on observable features appearing in transport phenomena. We present a systematic study showing on general grounds that the emitted distribution of carriers is not thermal.

This remains true even in the extreme situation where the defect is mascroscopically large and regions deep within the defect are well described by thermal ensembles. This effect can be interpreted as a generalization of the famous *Boundary Thermal Resistance* (BTR) [58] for contact points between two separated phases, featuring a discontinuity in the temperature profile [59–61]. Effects of interfaces have been recently addressed in the quantum case joining together two different integrable spin chains [62]. In the case under scrutiny, far from the defect the state is well described by a Generalized Gibbs Ensemble (GGE) [6,63] $e^{-\sum_j \beta_j \mathcal{Q}_j}$ built on the conserved charges $\mathcal{Q}_j$ of the integrable Hamiltonian. On the contrary, in the case of extended defects, the center of the interacting region relaxes to a thermal ensemble $e^{-\beta(H-\mu N)}$. We will show that between these two regimes there exists a finite-size interpolating region which survives even when the defect is infinitely extended. We call this effect Boundary Generalized Resistance (BGR) (see Fig. 1). Our claim is based on extensive numerical simulations, physical arguments and on Boltzmann-kinetic equations for weakly interacting, but extended defects.

## 2 The model: classical interacting fields on a lattice

Let $\psi_x$ be a classical complex field with Poisson brackets $\{\psi_x, \psi_{x'}^*\} = i\delta_{x,x'}$. For the sake of simplicity, we choose the bulk Hamiltonian to be noninteracting and hence diagonal in the Fourier space $\psi(k) = \sum_x e^{ikx}\psi_x$

$$H_{\text{bulk}} = \int_{-\pi}^{\pi} \frac{dk}{2\pi} E(k)\psi^*(k)\psi(k), \tag{1}$$

with $E(k)$ the dispersion law, which will be specified later on. The defect is interacting and in the form

$$V = \lambda \sum_x D(x/L)|\psi_x|^6. \tag{2}$$

The function $D(x)$ has compact support centered around $x = 0$ and encodes the spatial profile of the defect, while $\lambda$ and $L$ parametrize its strength and extension respectively. The choice of a $|\psi|^6$−interaction is motivated to avoid integrability also in the continuum limit, while in this case an interaction $\propto |\psi|^4$ would have been reduced to the integrable Non-Linear Schroedinger equation. Far from the defect, the state evolves according to the bulk Hamiltonian and in the late-time regime it locally equilibrates to the GGE $\langle \mathcal{O}(t,x) \rangle = \langle \mathcal{O} \rangle_{\text{GGE}(t,x)}$ described by a space-time dependent mode density $n_{t,x}(k)$. The latter obeys a simple kinetic equation

$$\partial_t n_{t,x}(k) + v(k)\partial_x n_{t,x}(k) = 0, \tag{3}$$

with $v(k) = \partial_k E(k)$ the group velocity. This equation can be derived from the time evolution of the two-point correlation function [64], which is directly connected to the mode density $\langle \psi_x^* \psi_{x'} \rangle = \int_{-\pi}^{\pi} \frac{dk}{2\pi} e^{ik(x-x')} n_{t,\frac{x+x'}{2}}(k)$. If $H_{\text{bulk}}$ is interacting, one can still write a kinetic equation within the framework of Generalized Hydrodynamics [16, 17], but its connection with local observables is less straightforward [65]; we thus opt to work with free systems.

At late times, the solution to the kinetic equation becomes self-similar $n_{t,x}(k) \to n_{\zeta=x/t}(k)$, with $\zeta$ being called the ray. Note that in this scaling limit any defect covering a finite domain ends up being shrunk to $\zeta = 0$ and sets the boundary conditions at $\zeta = 0^\pm$: the mode density of outgoing carriers is determined by ingoing carriers. Outgoing modes at $\zeta = 0^\pm$ are identified by the sign of the velocity $v(k) \lessgtr 0$, the ingoing one having opposite sign. In the (incorrect) assumption of thermalising defect, the outgoing carriers are thermally distributed

$$n_{\text{out}}^\pm(k) = \frac{1}{\beta_\pm(E(k) - \mu_\pm)}, \tag{4}$$

with some effective inverse temperature and chemical potential $(\beta_\pm, \mu_\pm)$, yet to be determined. Two constraints are found imposing the conservation of energy and density current across the defect and two more parameters need to be fixed employing some kinetic approach [62]. However, if parity symmetry around the origin holds, one can probe the very hypothesis of thermalising defect without introducing further assumptions, as $\beta_+ = \beta_- = \beta$ and $\mu_+ = \mu_- = \mu$, with $\beta, \mu$ determined by the incoming carriers ($q(k) = \{1, E(k)\}$ for the particle and energy current respectively)

$$\int_{v(k)>0} dk\, q(k)v(k)\left\{ n_{\text{in}}(k) - [\beta(E(k) - \mu)]^{-1} \right\} = 0. \tag{5}$$

**Preliminary numerical observations —** Probing the expanding lightcone in the scaling regime needs prohibitively long timescales and system sizes. These issues can be circumvented *i)* focusing on large, but finite systems encompassing the defect and *ii)* mimicking the infinitely long systems with suitable dissipative-driven boundaries. See Appendix A for further details. For reasons due to the forthcoming kinetic analysis, we choose a parabolic energy $E(k) = k^2$ for $k \in [-\pi, \pi]$ and periodically continued beyond the Brillouin zone, but other choices do not change the picture. In Fig. 2 a symmetric GGE is pumped into a box-shaped defect. Overall, we experienced that it is quite hard to observe noticeable drifts from the injected mode density for a wide choice of sizes and interactions. At fixed $L$ the deviations from the injected density are not monotonous in the interaction strength, since at $\lambda \to +\infty$ carries are purely reflected by an impenetrable barrier. On an intermediate scale of interactions, the outgoing carries deviate from the injected distribution, but they are far from being thermally distributed. This can be justified on the basis of kinetic considerations: for finite interactions, there is a non-zero probability that the moving carrier injected into the defect is reflected back after only a small number of scattering processes, not sufficient to make it to thermalise. This mechanism causes

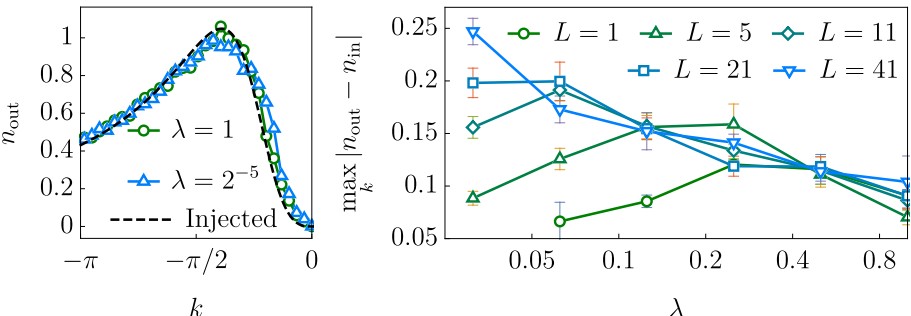

Figure 2: The defect is chosen in the form Eq. (2), having constant interaction $\lambda$ on $L$ sites and zero otherwise. We focus on the left edge of the defect and on the outgoing carrier $k < 0$. Left: the out distribution for different interactions is compared against the injected mode density $n_{\text{in}}$ (dashed line). Right: $\max_k |n_{\text{in}}(k) - n_{\text{out}}(k)|$ vs $\lambda$, for different sizes $L$.

carriers to be non-thermally distributed at the edges of the defect, and it is one of the possible mechanisms at the origin of the Boundary Generalized Resistance, as we depict in Fig. 1. The natural question we wish to address now is whether the BGR can be suppressed reducing such an effective barrier, by means of suitable choices of interactions and initial states. For finite interactions, low-energy carriers will be scattered back before than they can thermalise, pointing at the case of weak and extended defects as the most interesting regime (see also Fig. 2). This limit is amenable of an analytical analysis via a Boltzmann-kinetic equation, which nevertheless shows that the BGR still persists.

## 3 The Boltzmann scaling limit and the Boundary Generalized Resistance

In the limit of weak interactions, one locally describes the system with the mode density as if it was non-interacting, while the interactions scramble the mode density. The Boltzmann kinetic equation for weakly interacting systems is a textbook approach [66–69], nevertheless we provide a detailed derivation for completeness in Appendix B. In the presence of the defect, the Boltzmann kinetic equation is

$$\partial_t n_{t,x}(k) + v(k)\partial_x n_{t,x}(k) - \partial_x[\lambda D(x/L)U_x]\partial_k n_{t,x}(k) = \lambda^2 D^2(x/L)\mathcal{I}_k[n_{t,x}]. \tag{6}$$

Above, in addition to the ballistic propagation of carriers, one gets an effective potential $U = 18(\langle|\psi|^2_x\rangle)^2$ and a collision term $\mathcal{I}_k$, whose expression can be found in Appendix B, precisely in Eq. (B.11). We can finally motivate the choice for $E(k)$: the lattice dispersion law $E(k) = 2(1 - \cos k)$ is responsible of well-known divergences in $\mathcal{I}_k$ which needs to be properly regularized (see e.g. [68]), but can be avoided including more complicated hoppings [69]. Here, inspired by the continuum limit, we use the simplest choice $E(k) = k^2$. In the following, we are interested in the late-time physics, once the defect has reached a stationary state, hence we pose $\partial_t n = 0$. The effective potential $U$ acts as a repulsive barrier, hence carriers with energy smaller than $\sim \lambda$ will be reflected before they get the chance to interact, hence we wish to take $\lambda \to 0$. In order to have a non-trivial defect, we rescale its size in such a way $L\lambda^2$ is kept constant.

## 3.1 The scaling limit

We now carefully discuss the scaling limit that allows to reduce the effective barrier and enhance the role of the interactions. We start by describing the corrections to Eq. (6): the Botzmann equation we wrote is an expansion for small values of the interactions and in the derivatives of the mode density, hence it can feature corrections of the form $\mathcal{O}(\lambda^3)$, $\mathcal{O}(\lambda^2 \partial_x n_{t,x})$ and $\mathcal{O}(\lambda \partial_x^2 n_{t,x})$, plus higher order corrections. So far, we have not imposed any relation between the interaction $\lambda$ and the lengthscale $L$, hence each of the corrections enlisted above can dominate the others with a suitable choice of parameters.

As anticipated, we are interested in the limit of weak interactions $\lambda$, but large defect size $L$: below we give a more quantitiative discussion, showing the correct scaling indeed keeps $\lambda^2 L$ finite. In this limit, the corrections mentioned above become negligible and the Boltzmann equation further simplifies. We start by noticing that in the limit of small $\lambda$ at $L$ fixed, the effective potential $U_x$ is dominant over the collision integral. In this regime, $U_x$ acts as a potential barrier and low-energy carriers are immediately reflected, without having the time to relax: it is thus clear that in this limit, the scrambling effects of the interactions are not effective.

Hence, we require a limit where $U_x$ becomes negligible and $\mathcal{I}_k$ dominant. This situation can be achieved in the large $L$ limit. Let us imagine this is the case and that $U_x$ can be neglected: we now impose this condition self-consistently. If this is the case, then $n_{x,t}$ has a spatial inhomogeneity dictated by the interaction strength, thus $\partial_x n_{x,y} \sim \lambda^2$. The force term associated with the effective potential has now two contributions

$$\partial_x(\lambda D(x/L)U_x) = \lambda L^{-1}D'(x/L)U_x + \lambda D(x/L)\partial_x U_x\,, \tag{7}$$

with $D'$ being the derivative of $D$, which is assumed to be a smooth function. Since $U_x$ depends on the mode density and $\partial_x n_{x,t} \sim \lambda^2$, one has $\lambda D(x/L)\partial_x U_x \sim \lambda^3$, hence it is subleading with respect to the collision term. Besides, all the corrections to Eq. (6) become explicitly subleading with respect to the collision integral, hence they can be neglected. The first term in Eq. (7) accounts for the explicit spatial inhomogeneity of $D$. One can neglect this contribution with respect to the collision integral if

$$L^{-1}\lambda \ll \lambda^2 \quad \Rightarrow \quad L \gg \lambda^{-1}\,. \tag{8}$$

If this condition holds, in the $\lambda \to 0$ limit we reach the simplified stationary Boltzmann equation

$$v(k)\partial_x n_x(k) = \lambda^2 D^2(x/L)\mathcal{I}[n_x(k)]\,. \tag{9}$$

In order to make the role of the interaction explicit, we perform a change of variable

$$X = \frac{\int_{-\infty}^{x} \mathrm{d}y\, [D(y/L)]^2}{\int_{-\infty}^{\infty} \mathrm{d}y\, [D(y/L)]^2} \tag{10}$$

and define the effective interaction

$$\Lambda \equiv L\lambda^2 \int_{-\infty}^{\infty} \mathrm{d}x\, [D(x)]^2\,, \tag{11}$$

where the scaling with $L\lambda^2$ is made explicit. In these new coordinates, the defect is supported on the interval $X = [0,1]$ and the stationary Boltzmann equation reads

$$v(k)\partial_X n_X(k) = \Lambda \mathcal{I}_k[n_X]\,, \tag{12}$$

where the boundary conditions at $X = \{0,1\}$ are set by the injected mode density.

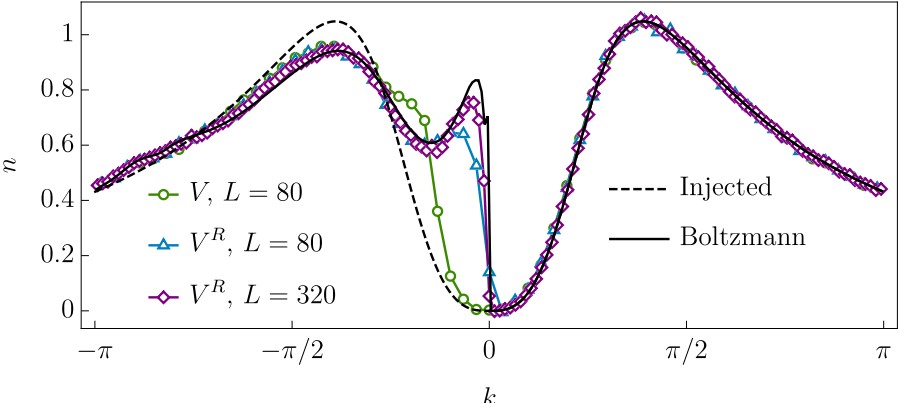

Figure 3: The Boltzmann scaling (12) is compared to microscopic simulations (markers) for the mode density on the left of the defect. $D(x)$ is a smoothed step function (see Appendix C); while changing $L$, the microscopic interaction $\lambda$ is adjusted to keep $\Lambda = L\lambda^2 = 0.05$. The injected mode density (dashed line) at $k > 0$ is fixed, while the carriers leaving the interacting region approach the Boltzmann result as $L \to \infty$. $V^R$ is obtained setting $A = -7.84$ (see main text).

If $\Lambda$ is small, the collisions will essentially not affect the mode density and let it propagate across the defect unchanged. In the opposite regime, the defect is strongly interacting and mixes the momenta. We are of course interested in the second case. Notice that $\Lambda$ can be made large while fulfilling both the requirements of small interaction $\lambda \ll 1$ and Eq. (8). Actually, keeping $\Lambda$ constant and taking $\lambda \to 0$ implies Eq. (8).

The computation of the collision term is extremely demanding due to the presence of multidimensional integrals; here, we devised a new numerical algorithm which significantly reduces the computational cost (see Appendix C), allowing us to tackle Eq. (12). In Fig. 3, we benchmark the Boltzmann kinetic equation against microscopic simulations, finding excellent agreement. At infinite $L$, the effective potential becomes irrelevant and is indeed absent from Eq. (12). However, the approach to the scaling limit is rather slow in practice $\sim L^{-1/4}$, as we now discuss.

Let us fix the effective interaction $\Lambda$ and use as scaling parameter the defect size $L$: we now go back to Eq. (6) and analyze the effect of the effective potential $U$. The reasoning is as follows: the effective potential acts as a barrier for the incoming carriers and those that do not have enough energy to overcome it are reflected. These reflected excitations are responsible of a drift towards the Boltzmann prediction that goes as $\sim L^{-1/4}$. More precisely, we expect a carrier of momentum $k$ to be reflected if its energy is less than the energy barrier $E(k) < \lambda \max_x U_x$. Using that $E(k) = k^2$ and assuming the normalization $\int_{-\infty}^{\infty} \mathrm{d}x \, [D(x)]^2 = 1$ for simplicity, we get that momenta such that

$$|k| < L^{-1/4} \Lambda^{1/4} \sqrt{\max_x U_x} \qquad (13)$$

are reflected and do not experience any scrambling. Hence, in order to help the slow convergence in $L$, in Fig. 3 we reduced the effect of $\max_x U_x$, while keeping the same $\mathcal{I}_k$. Indeed, as we show in Appendix B, a renormalization of the interaction (2) $V \to V^R = \lambda \sum_x D(x/L)\big(|\psi_x|^6 + A|\psi_x|^2\big)$ leaves $\mathcal{I}_k$ and the scaling to (12) unscathed, but the effective potential in (6) gets renormalized as $U \to U^R = 18(\langle|\psi|_x^2\rangle)^2 + A)$: an appropriate choice of $A$ quickens the convergence to the Boltzmann scaling.

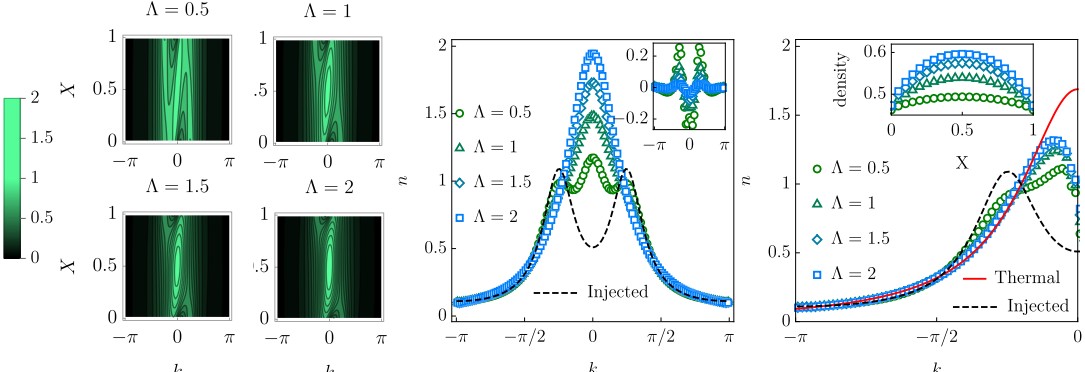

Figure 4: By numerically solving Eq. (12), we explore the defect in the scaling regime for different interactions $\Lambda$ with a constant injected mode density (central panel, dashed line). Left: density plot of the phase-space density across the defect. As $\Lambda$ is increased, the injected peaks are merged into a single central peak that approaches the thermal distribution. Center: mode density at $X = 0.5$. Inset: relative distance from a thermal fit; for $\Lambda = 2$, $n(k)$ is almost indistinguishable from a thermal distribution. Right: outgoing mode density at $X = 0$ compared with the injected distribution (dashed) and the thermal prediction (continuous red line). The thermal prediction fails even for large interactions, despite the center of the defect being essentially thermal. Inset: density profile across the defect. For large interactions, the system approaches a homogeneous state in the middle of the defect, with a manifest BGR at the boundaries.

Once the validity of the method has been assessed, we explore sizes of the defect unreachable with microscopic numerical simulations. Intuitively, large values of $\Lambda$ describe an extended defect that will eventually behave as a thermodynamic system on its own, thus thermalising in its center. This is indeed shown in Fig. 4 where we explored the phase space across the defect: for large $\Lambda$, the mode density becomes approximately homogeneous in the deep bulk and is well-fitted by a thermal distribution. Nevertheless, the BGR takes place: carriers around the edges remain out-of-equilibrium and the outgoing momentum distribution is clearly non-thermal, despite thermal equilibrium has been attained at the center.

**The linearized regime —** It is natural to investigate the response of the defect to small perturbations around homogeneous thermal states $n_{t,x}(k) = n_{\text{th}}(k) + \epsilon \delta n_{t,x}(k)$ with $\epsilon$ small. In the scaling limit, the homogeneous thermal state $n_{\text{th}}(k)$ is obviously a stationary solution of Eq. (12) and one can study weak deviations from it. In this regime, the collision integral in Eq. (12) is linearized and is computed only once, so that one can access much larger defects (see Appendix C), confirming the existence of the BGR. Taking advantage of the linearity of the problem, in Fig. 5 we injected an asymmetric perturbation $\delta n(k) = \sin(k)$ for $k > 0$ and $\delta n(k) = 0$ for $k < 0$ over a thermal state with $\beta = \mu = 1$. For large interactions, the transmitted part vanishes (inset) proving that in the bulk of a large defect one relaxes to the unperturbed thermal state. In this limit, all carriers are reflected back, but their distribution remains non-thermal.

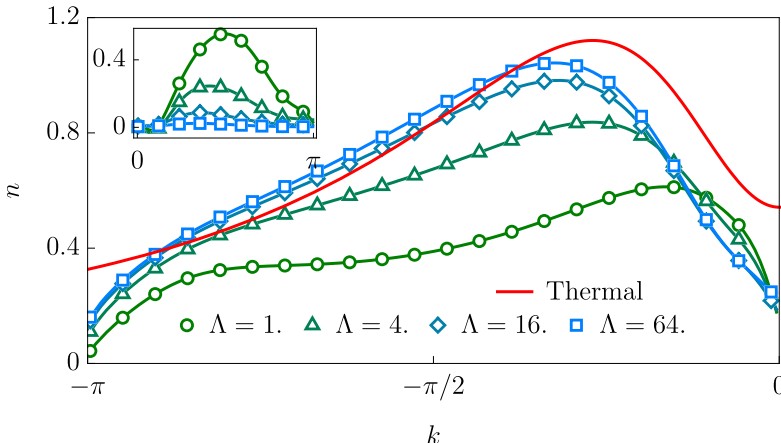

Figure 5: Reflected mode density obtained with the linearized Boltzmann equation (see main text for parameters). Inset: transmitted mode density. For large effective interactions $\Lambda$, all the carriers are reflected back.

## 4  Conclusions

In this work, we investigated the scrambling effects of thermalising mesoscopic impurities embedded in a non-thermalising environment. We show how carriers flowing out from the defect are in general not thermally distributed even in the extreme case of extended defects which thermalise in the center. This is due to a generalization of the Boundary Thermal Resistance, taking place at the interface between the defect and the bulk of the system. In this work, we focus on a classical model of interacting fields on a lattice: this allowed us to perform large-scale numerical simulations far beyond the capability of the state-of-the-art numerical quantum algorithms. However, the underlying mechanism can be framed within a Boltzmann-kinetic equation whose applicability can be extended to quantum systems [68,69]. Hence, the Boundary Generalized Resistance is envisaged to take place in quantum setups as well. Several questions remain open for the future. First, it would be useful to formulate the BGR at a junction, with a minimal set of phenomenological parameters ($\sim$ generalised resistances) which encode the transport properties of multiple conserved quantities, without the need to fully solve the dynamics. Secondly, here, we focused on the case where the bulk Hamiltonian is free and interacting-integrable bulk Hamiltonians are a natural next step to be addressed: the study of collision terms in the framework of Generalized Hydrodynamics is still at its infancy [70–72], but an analysis in the same spirit of our kinetic equation can be envisaged. Beside integrability, there are several ways to hinder thermalisation while retaining non-trivial transport, such as Hilbert space fragmentation [9,73] and it is natural to wonder about interfaces between fragmenting and non-fragmenting Hamiltonians.

**Funding information**  AB acknowledges support from the Deutsche Forschungsgemeinschaft (DFG, German Research Foundation) under Germany's Excellence Strategy-EXC-2111-390814868.

# A  The microscopic simulations: mimicking infinite systems with dissipative-driven boundaries

As we anticipated in the main text, while focusing on classical systems allows us to explore much longer times and sizes when compared with the quantum case, probing mesoscopic defects appears rather challenging. Simulating large systems with open or periodic boundary conditions sets a maximum timescale, after which carriers leaving the defect will hit the boundary and then come back to the defect. In order to avoid this effect, we simulate the infinitely large systems by means of suitable driven-dissipative boundaries. The physical picture behind this method is the following: in the bulk and far from the defect, carriers at different momenta are non-interacting and can be thus independently split into ingoing and outgoing carrier. By a convenient choice of driving and dissipation one can *i)* remove the outgoing carriers from the system in such a way they do not come back to the defect and *ii)* inject a tunable distribution of carriers which remains constant in time. Enforcing these two points, we trustfully simulate the action of an infinitely extended system on the defect region. Hereafter, we discuss the details of such an implementation and explain how, by tuning the driving, we can shape the GGE distribution of the injected carriers at will. Let us consider the following equation of motion

$$i\partial_t \psi_x = \left( \sum_{x'} A_{x-x'} \psi_{x'} \right) + \lambda D(x/L) |\psi_x|^2 \psi_x - \gamma_x \left[ i\psi_x + \xi_x(t) \right], \tag{A.1}$$

where $A_x$ is such that $E(k) = \sum_x e^{ikx} A_x$, while $\gamma_x i\psi_x$ is responsible for dissipative dynamics (it makes the time evolution non-unitary) and $\xi_x(t)$ is a gaussian random noise with zero mean and variance

$$\langle \xi_{t,x}^* \xi_{t',x'} \rangle = \delta(t-t') \int \frac{dk}{2\pi} e^{ik(x-x')} s(k). \tag{A.2}$$

In the absence of dissipation and noise, this equation can be derived from the Hamiltonian (1) (2). We choose $\gamma_x$ to be weakly inhomogeneous, in such a way it is absent on the defect. Carriers traveling across the dissipative region decay on a time scale $\sim 1/\gamma_x$. We tune the dissipation in such a way they decay before reaching the edges of the system, never coming back to the defect. On the contrary, the external drive creates excitations in a controlled way, allowing for a determination of the injected mode density. More specifically, we engineer the following setup

1. The defect covers a region of size $L$ centered around zero, namely $[-L/2, L/2]$.

2. The whole system has length $\bar{L} \geq L$ and covers the region $[-\bar{L}/2, \bar{L}/2]$. We impose periodic boundary conditions on the whole system, but other boundary conditions are equivalent. $\bar{L}$ must be larger than $L$, but it is not needed $\bar{L} \gg L$. The conditions on $\bar{L}$ will be clear soon.

3. We choose $\gamma_x$ to be a smooth positive function, acting non trivially in the regions $[-\bar{L}/2, -\bar{L}/2 + b]$ and $[\bar{L}/2 - b, \bar{L}/2]$ and zero otherwise. The size of the boundary region $b$ is large, but we keep the dissipative boundaries well separate from the defect. In particular, we wish to keep an extended region evolving without dissipation neither defect. This region is used to numerically compute the mode density distribution, as clarified below.

4. In the limit of weak and smooth $\gamma_x$, one can describe the boundaries with a kinetic equation

$$\partial_t n_{t,x}(k) + \partial_x (v(k) n_{t,x}(k)) = -2\gamma_x n_{t,x}(k) + \gamma_x s(k), \tag{A.3}$$

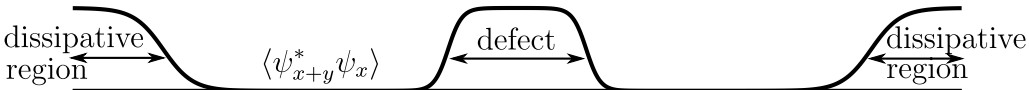

Figure 6: Sketch of the setup used for the microscopic simulations.

which is valid far from the defect and whose derivation is postponed. The stationary case $\partial_t n_{t,x}(k) = 0$ can be easily numerically found taking advantage of the diagonality in the $k$ space and the mode density injected on the defect determined as a function of $s(k)$. Tuning $s(k)$, one can change the distribution of the carriers injected into the defect.

5. From a zero field configuration $\psi_x = 0$, we let the system evolve with the equation of motions (A.1) and we follow the time evolution of several observables. We evolve for very long times until we are sure a stationary state is reached. At this point, we start sampling the observables of interest: we keep on time-evolving the field configuration and consider the time average of the observables, taking advantage that the system is self averaging once the steady state is attained. In particular, the mode density is obtained locally computing $\langle \psi_{x+y}^* \psi_x \rangle$ with $x$ far from both the dissipative boundaries and the defect. Then, the Fourier transform of the correlator is taken, obtaining the mode density.

In Fig. 6, we provide a sketch of the setup. We finally discuss a quick derivation of Eq. (A.3). For the sake of simplicity, we derive the effect of the dissipation in the homogeneous case: the (weak)inhomogeneous case is then recovered promoting $\gamma$ to be inhomogeneous and adding the gradient term in (A.3). The microscopic equation of motion is diagonal in the momentum space $i\partial_t \psi(k) = E(k)\psi(k) - i\gamma\psi(k) + \gamma\xi(t,k)$ with $\psi(k)$ and $\xi(t,k)$ be the Fourier transform of $\psi_x$ and $\xi_x(t)$ respectively. Then, the mode density is defined from the correlation of the fields $\langle \psi^*(k)\psi(q) \rangle = \delta(k-q)n(k)$: computing its time derivative from the microscopic equations one easily gets $\partial_t n(k) = -2\gamma n(k) + \gamma s(k)$, which is promoted to Eq. (A.3) once $\gamma$ is made weakly inhomogeneous.

# B  Derivation of the Boltzmann equation

The derivation of the Boltzmann kinetic equation for weakly interacting models recurs in several instances in the literature, see e.g. Refs. [66–69]. Here, we wish to provide the expression for the collision intergral and a quick derivation. Besides, for the sake of simplicity, since we focus on the collision integral we can work in the homogeneous case. Then, the kinetic equation can be promoted to be inhomogeneous adding the proper gradient terms. For the derivation, we follow [69] and the strategy is summarized here below:

1. We write the equation of motion for the two point correlator in Fourier space, from which we can extract the mode density. Since the equations are non linear, the time derivative of the two point correlator couples to higher order correlators. Nevertheless, in the weak interacting limit these objects are small $\mathcal{O}(\lambda)$.

2. We write the equation of motion for higher point correlators and we truncate them to the first order in $\lambda$. As we will see, this truncation amounts to consider only two-point correlators in the equation of motion. Hence, these equations can be solved and fed into the equations of the previous point, resulting in a Boltzmann equation which is closed for the two point correlator (or equivalently mode density).

We consider Hamiltonians in the following form

$$H = \sum_{x\,x'} A_{x-x'} \psi_x^* \psi_{x'} + \lambda \sum_x \mathcal{V}(|\psi_x|^2), \tag{B.1}$$

where we recall $A_x$ is real and $E(k) = \sum_x e^{ikx} A_x$ and the interaction is a power-expandable function

$$\mathcal{V}(x) = \sum_\ell \frac{c_\ell}{\ell!} x^\ell, \tag{B.2}$$

with proper coefficients $c_\ell$. In the main text, we focus on the case where $c_\ell$ is non zero for $\ell = 3$ and vanishes otherwise, but we wish to address the more general case. We now write the equations of motion for $\langle \psi_x^* \psi_{x'} \rangle$

$$i\partial_t \langle \psi_x^* \psi_{x'} \rangle = -\lambda \langle \mathcal{V}'(|\psi_x|^2) \psi_x^* \psi_x \psi_{x'} \rangle + \lambda \langle \psi_x^* \mathcal{V}'(|\psi_{x'}|^2) \psi_{x'}^* \psi_{x'} \rangle. \tag{B.3}$$

We explicitly use translational invariance which cancels terms in the form $\sum'_{x'} A_{x'-x''} \langle \psi_x^* \psi_{x''} \rangle$ on the right hand side. Above, $\mathcal{V}'(x) = \partial_x \mathcal{V}(x)$. On the right hand side, multipoint correlators appear, hence we now consider the equation of motion for these objects

$$i\partial_t \langle \prod_i \psi_{x_i}^* \prod_i \psi_{y_i} \rangle = \sum_{i'} \sum_z A_{x_{i'}-z} \langle \psi_z^* \prod_{i\neq i'} \psi_{x_i}^* \prod_i \psi_{y_i} \rangle - \sum_{i'} \sum_z A_{y_{i'}-z} \langle \prod_i \psi_{x_i}^* \psi_z \prod_{i\neq i'} \psi_{y_i} \rangle$$

$$\lambda \sum_\ell \frac{c_{\ell+1}}{\ell!} \left[ \sum_{i'} \langle \prod_i \psi_{x_i}^* |\psi_{y_{i'}}|^{2\ell} \psi_{y_{i'}} \prod_{i\neq i'} \psi_{y_i} \rangle - \sum_{i'} \langle |\psi_{x_{i'}}|^{2\ell} \psi_{x_{i'}}^* \prod_{i\neq i'} \psi_{x_i}^* \prod_i \psi_{y_i} \rangle \right]. \tag{B.4}$$

So far, no approximation has been made. As a next step, we divide the correlator in its connected parts and it is immediate to notice the general structure $\partial_t \langle ... \rangle_c = \lambda[\text{gaussian part}] + \mathcal{O}(\lambda^2)$. Hence, at the price of neglecting $\mathcal{O}(\lambda^2)$ terms we truncate at the gaussian level. Afterwards, once the multipoint connected correlator is plug in Eq. (B.3) we will get a $\propto \lambda^2$ term plus $\mathcal{O}(\lambda^3)$ neglected corrections, due to the already present $\lambda$ prefactor in Eq. (B.3). Focusing on the connected part of Eq. (B.4) and using the mentioned truncation, one gets

$$i\partial_t \langle \prod_i \psi_{x_i}^* \prod_i \psi_{y_i} \rangle_c =$$

$$\sum_{i'} \sum_z A_{x_{i'}-z} \langle \psi_z^* \prod_{i\neq i'} \psi_{x_i}^* \prod_i \psi_{y_i} \rangle_c - \sum_{i'} \sum_z A_{y_{i'}-z} \langle \prod_i \psi_{x_i}^* \psi_z \prod_{i\neq i'} \psi_{y_i} \rangle_c$$

$$+ \lambda \sum_\ell c_{\ell+1} \frac{(\ell+1)! \langle |\psi|^2 \rangle^{\ell-n+1}}{(\ell-n+1)!} \left[ \sum_{i'} \prod_i \langle \psi_{y_i}^* \psi_{x_{i'}} \rangle \prod_{i\neq i'} \langle \psi_{x_{i'}}^* \psi_{x_i} \rangle - \right.$$

$$\left. \sum_{i'} \prod_i \langle \psi_{y_{i'}}^* \psi_{x_i} \rangle \prod_{i\neq i'} \langle \psi_{y_i}^* \psi_{y_{i'}} \rangle \right]. \tag{B.5}$$

Above, $\langle |\psi|^2 \rangle \equiv \langle |\psi_x|^2 \rangle$ for any point due to translational invariance and acts as a renormalization of the interaction. This equation is better expressed in the Fourier space, where the two point correlator becomes the mode density $\langle \psi_x^* \psi_y \rangle = \int \frac{dk}{2\pi} e^{ik(x-y)} n(k)$. We define the multipoint connected correlator in the momentum space as

$$\langle \prod_{i=1}^n \psi_{x_i}^* \prod_{i=1}^n \psi_{y_i} \rangle_c = \int \frac{d^n k}{(2\pi)^n} \frac{d^n q}{(2\pi)^n} e^{i\sum_i k_i x_i - i\sum_i q_i y_i} C(\{k_i\}|\{q_i\}) 2\pi\delta\left(\sum_i k_i - \sum_i q_i\right). \tag{B.6}$$

The Dirac delta in the momentum space ensures the translational invariance and must be interpreted modulus $2\pi$, since we have a finite Brillouin zone. Eq. (B.5) is then rewritten in the momentum space as

$$i\partial_t C(\{k_i\}|\{q_i\}) = \left[\sum_i (E(q_i) - E(k_i))\right] C(\{k_i\}|\{q_i\})$$
$$+ \sum_\ell c_{\ell+1} \frac{(\ell+1)! \langle|\psi|^2\rangle^{\ell-n+1}}{(\ell-n+1)!} \prod_i n(k_i)n(q_i) \sum_i \left(\frac{1}{n(q_i)} - \frac{1}{n(k_i)}\right). \quad (B.7)$$

We now time-integrate these equations in the following approximation. The mode density $n$ evolves on a $\sim \lambda^{-1}$ time scale, as it is clear from (B.3). This time scale is much larger than the dephasing time scale set by the energy $E(k)$, hence in the limit of weak interaction we can time integrate the above equation as if $n(k)$ was constant in time, which results in

$$C(\{k_i\}|\{q_i\}) =$$
$$- \lambda \left(\sum_\ell c_{\ell+1} \frac{(\ell+1)! \langle|\psi|^2\rangle^{\ell-n+1}}{(\ell-n+1)!}\right) \frac{\prod_i n(k_i)n(q_i) \sum_i \left([n(q_i)]^{-1} - [n(k_i)]^{-1}\right)}{\sum_i E(q_i) - \sum_i E(k_i) - i0^+}. \quad (B.8)$$

As a final step, this expression is plug into Eq. (B.3) and everything is expressed in the momentum space, where the Boltzmann equation is most clearly written. In doing so, one should pay attention that in Eq. (B.3) there are multipoint correlators, while $C(\{k_i\}|\{q_i\})$ is only the connected part. Building on the fact that connected correlator are order $\lambda$, we can use the truncation

$$\langle(|\psi_x|^{2\ell})\psi_x\psi_{x'}\rangle = \text{(gaussian part)} + \sum_a \binom{\ell+1}{a}\binom{\ell}{a}a!\langle|\psi|^2\rangle^a \langle(\psi_x^\dagger)^{\ell+1-a}\psi_x^{\ell-a}\psi_{x'}\rangle_c + \dots. \quad (B.9)$$

The gaussian part in the above does not contribute to Eq. (B.3) and after a rearrangement of the terms one finally gets

$$\partial_t n(k) = \lambda^2 \mathcal{I}_k[n], \quad (B.10)$$

with

$$\mathcal{I}_k[n] = \lambda^2 \sum_\ell \mathcal{C}_\ell(|\psi|^2) \int \frac{d^{\ell-1}k}{(2\pi)^{\ell-1}} \frac{d^\ell q}{(2\pi)^\ell} 2\pi\delta\left(\sum_i k_i - \sum_i q_i\right) 2\pi\delta\left(\sum_i E(q_i) - \sum_i E(k_i)\right) \times$$
$$\prod_{i=1}^\ell n(k_i)n(q_i) \sum_{i=1}^\ell \left(\frac{1}{n(k_i)} - \frac{1}{n(q_i)}\right)\Bigg|_{k_1=k}, \quad (B.11)$$

where the coefficients $\mathcal{C}_\ell$ are (we recall that $c_i$ are the Taylor coefficient of the interaction (B.2))

$$\mathcal{C}_\ell(|\psi|^2) = \sum_{a,s} c_{\ell+a} c_s \frac{(\ell+a)!}{a!\ell!} \frac{s!}{(\ell-1)!(s-\ell)!} (\langle|\psi|^2\rangle)^{s-\ell+a}. \quad (B.12)$$

This concludes the derivation of the collision integral $\mathcal{I}_k[n]$. We notice that it can be easily checked that the total number of particles and energy, $\int \frac{dk}{2\pi} n(k)$ and $\int \frac{dk}{2\pi} E(k)n(k)$ respectively, are conserved by Eq. (B.10). Moreover, thermal states $n(k) = [\beta(E(k) - \mu)]^{-1}$ are stationary solutions for any temperature and chemical potential, as it should be.

It should be stressed that the presence of multidimensional integrals in Eq. (B.11) is a mayor bottleneck in its numerical evaluation, especially for high dimensionality. The integral can be performed by means of Metropolis methods (see eg. Ref. [71]), but this approach is

very costly already in the homogeneous case, hence the inhomogeneous Boltzmann equation (where the collision integral must be computed at each point on the space grid) seems out of reach. In order to tackle these technical difficulties, we devised a new algorithm presented in the next section, which can efficiently compute $\mathcal{I}_k$. Most importantly, its complexity remains constant increasing the dimensionality of the integrals appearing in Eq. (B.11), making it very suited also to access many-body interactions.

## C  The numerical solution of the Boltzmann equation

Our main interest resides in finding the stationary solution of the inhomogeneous Boltzmann equation in the scaling limit (12). In order to do so, we see Eq. (12) as the stationary state of

$$\partial_t n_X(k) + v(k)\partial_X n_X(k) = \Lambda \mathcal{I}_k[n_X]. \tag{C.1}$$

Hence, we numerically solve the above equation until a stationary solution is reached. We discretize the space in Eq. (C.1) in an uniform grid $\{X_i = \mathrm{d}X(i - 1/2)\}_{i=1}^N$ with spacing $\mathrm{d}X = 1/(N+1)$ and the derivative is Eq. (C.1) is discretized with left or right increments depending on the sign of the velocity

$$\partial_t n_{X_i}(k) + v(k)\left[\theta(v(k))\frac{n_{X_i}(k) - n_{X_{i-1}}(k)}{\mathrm{d}X} + \theta(-v(k))\frac{n_{X_{i+1}}(k) - n_{X_i}(k)}{\mathrm{d}X}\right] = \Lambda \mathcal{I}_k[n_{X_i}]. \tag{C.2}$$

Above, $\theta(x)$ is the Heaviside Theta function and the mode density in $X_0$ and $X_{N+1}$ is fixed by the injected mode density $n_{X_0}(k) = n_{X_{N+1}}(k) = n_{\mathrm{in}}(k)$. Notice that the velocity-dependent discretization of the derivatives ensures the correct coupling with the boundary conditions. This spatial discretization is then also further discretized in the momentum space on an uniform grid, then the time evolution is Trotterized with a finite $\mathrm{d}t$ and the mode density is alternatively evolved with the kinetic term and with the collision integral

$$n'_{X_i}(t, k) = n_{X_i}(t, k) - \mathrm{d}t\, v(k)\left[\theta(v(k))\frac{n_{X_i}(t, k) - n_{X_{i-1}}(t, k)}{\mathrm{d}X}\right.$$
$$\left. + \theta(-v(k))\frac{n_{X_{i+1}}(t, k) - n_{X_i}(t, k)}{\mathrm{d}X}\right], \tag{C.3}$$

$$n_{X_i}(t + \mathrm{d}t, k) = n'_{X_i}(t, k) + \mathrm{d}t\, \Lambda\, \mathcal{I}_k[n'_{X_i}(t)]. \tag{C.4}$$

The stability of the algorithm requires $\mathrm{d}t < \mathrm{d}X/\max_k |v(k)|$. With the choice $E(k) = k^2$ one of course has $\max_k |v(k)| = 2\pi$. We are now left out with the challenging task of computing $\mathcal{I}_k$. This can be greatly simplified looking at $\mathcal{I}_k$ in the Fourier space. We define

$$\tilde{\mathcal{I}}_j = \int_{-\pi}^{\pi} \frac{\mathrm{d}k}{2\pi} e^{ikj} \mathcal{I}_k. \tag{C.5}$$

In the same spirit, we define the auxiliary functions

$$F_j(\tau) = \int \frac{\mathrm{d}p}{2\pi} e^{ijp + i\tau E(p)} n(p) \qquad G_j(\tau) = \int \frac{\mathrm{d}k}{2\pi} e^{ijp + i\tau E(k)}, \tag{C.6}$$

where $\tau$ plays the role of an auxiliary time. Then, it is a simple exercise to see that Eq.(B.11) can be rewritten as

$$\tilde{\mathcal{I}}_j = \int_{-\infty}^{\infty} \mathrm{d}\tau \sum_{j'=-\infty}^{\infty} \sum_{\ell=1}^{\infty} \mathcal{C}_\ell(|\psi|^2) |F_{j'}(\tau)|^{2(\ell-1)} \times$$
$$\left[(\ell - 1)\frac{F_{j'}^*(\tau)G_{j'}(\tau)}{F_{j'}(\tau)} F_{j'-j}(\tau) + F_{j'}^*(\tau)G_{j'-j}(\tau) - \ell F_{j'-j}(\tau)G_{j'}^*(\tau)\right]. \tag{C.7}$$

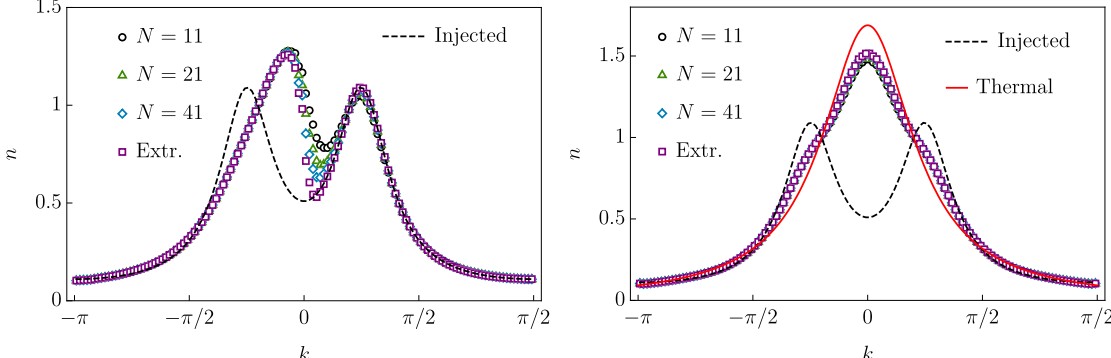

Figure 7: We provide the data for different spatial discretizations in one of the cases analyzed in Fig. 4, focusing on the example $\Lambda = 1$. Left: we plot the mode density on the first site of the discretization. Right: we focus on the central site. Symbols are the numerical solution of the stationary state for different number of sites $N$ in the spatial discretization and the extrapolated data. The dashed line is the injected mode density, the red continuum line on the right panel is the thermal fit based on the extrapolated data. See the text for further discussion.

Hence, the multidimensional integral has been converted in a sort of auxiliary quantum mechanical problem, where the wavefunctions $F_j$ and $G_j$ live on an infinite lattice and the collision integral is obtained by the time integration of a non-linear observable. Hence, this amounts to a two-dimensional integration, regardless the dimensionality of the integrals in the momentum space, with a large boost in efficiency. In practice, we proceed as it follows. We pick two large integers $M \gg \tilde{M}$, the auxiliary system lives on a lattice $[-M/2, M/2]$ and periodic boundary conditions are assumed. For $\tau = 0$, $G_j(0) = \delta_{j,0}$, while $F_j(0)$ is numerically computed from the mode density $n(k)$, then $F_j$ is truncated in such a way $F_{|j|>\tilde{M}/2} = 0$. This procedure keeps $n(k)$ smooth during the real time evolution. Then, the wavefunctions $F_j$ and $G_j$ are evolved in the auxiliary time $\tau$ in steps $d\tau$ and the value of $\tilde{\mathcal{I}}_j$ is updated. The auxiliary time evolution proceeds until a maximum cutoff $T$, which is set by the system's size. In practice, one needs $M > \tilde{M} + 2 \max_k |v(k)| T$. Overall, the computational cost of computing $\mathcal{I}_j$ scales as $\propto (d\tau)^{-1} T M \log M$: the $\sim M \log M$ scaling is due to the fact we use a fast Fourier transform to go back and forth from $p$ to $j$ space to compute the time evolution of $F_j$ and $G_j$, as well as the convolution in Eq. (C.7).

The values of the discretizations and truncations are adjusted until convergence is attained. The algorithm conserves the particle density up to machine precision, while the energy conservation depends on the choice of the parameters. For the simulations, we used $\tilde{M} = 2^8$, $M = 2^{12}$, $T = 130$ and $d\tau = 0.03$ and the momenta are discretized on a grid of $2^8$ points. With this choice the energy is conserved up to $5 \times 10^{-4}$. For what concerns the spatial discretization in Eq. (C.2), this largely depends whether we are tackling the full Boltzmann equation or the linearized version. Indeed, in the second case the linearized collision integral is computed once for all with the same methods, the large matrix is stored and then used for the time evolution. This allows us to consider very fine spatial discretizations (up to 200 points) and easily attain convergence. In the non-linear case, the collision integral must be computed at each time step and for each spatial point: this is the most costly part. Hence, in this case we consider $\{X_i\}_{i=1}^N$ with $N = \{11, 21, 41\}$ and then extrapolate. In Fig. 7 we focus on one example among the cases provided in Fig. 4, namely the one with interaction $\Lambda = 1$. On the left panel we provide the profile of the mode density on the first site of the discretization $X_1$ for the case $N = \{11, 21, 41\}$ and then the extrapolated value. We used a quadratic extrapolation

$1/N$ up to the quadratic order. For $k < 0$, the carriers are leaving the defect and are of central interest for Fig. 4. For $k > 0$ the carriers have been just injected on the defect and, by continuity, they should be described by the injected mode density (dashed line). We experience slow convergence in particular for small positive momenta. This is expected, since we are very far from equilibrium, hence the collision term gives important contributions. Looking at the kinetic equation $v(k)\partial_x n = \mathcal{I}_k[n]$, one immediately see that small momenta s.t. $v(k) \sim 0$ have bigger gradients in the mode density, hence a slower convergence in the spatial discretization is expected. However, the quadratic extrapolation well captures the $k > 0$ behavior, which supports the correctness of the extrapolation also in the $k < 0$ case. In the right panel we provide the mode density in the center of the defect. Here, the interactions had time to bring the mode density closer to the thermal state (albeit there is still a clear distinction), hence the collision integral has a smaller contribution. As a consequence, the gradient in the mode density is reduced and the convergence in the spatial discretization enhanced.

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
