# Peer review of "Transport through interacting defects and lack of thermalisation"

_SciPost Physics, doi:SciPost Phys. 12, 060 (2022)_

## Round 2 · Referee Report · Anonymous · 2021-10-27

Strengths

1- Nice trick to treat infinite systems numerically
2- Nice results on the effects of interaction on the thermalization properties

Weaknesses

1- Lack of clarity on the characteristics of the model studied and the implications of the results on the field

Report

This is a nice and original work on a classical 1-d model written as if it were a paper on a quantum model, which makes reading it quite confusing. It considers a *classical* system of quadratic oscillators with a non-linear potential confined in a region of size L.
The authors observe how thermalization is achieved in such a system as a function of the size L and the non-linearity strength $\lambda$. They open the system via dissipative-driven boundaries, which are supposed to mimic infinite systems in the numerics, which is the classical analog of having a Lindbladian superoperator on the extreme left and right sites of a quantum spin chain. They observe remnants of integrability in transport properties even when the bulk of the system is instead described by a thermal statistics. This is interpreted as a generalized boundary thermal resistance effect. I am not an expert on this last phenomenon but the interpretation seems correct. The paper deserves publication, however the authors need to rewrite big chunks of the paper to make it clear the characteristics of the system they are considering. If one were to read the abstract and skip to Eq.(1), one would think this is a work on an integrable *quantum* model (as I did after a first glance!). They even use the notation $\psi^\dagger$ in Eq.(1) and they talk about Wigner function shortly after. They also talk about "quasiparticles" while these are just the modes of the unperturbed, classical, quadratic oscillators. Written like this, the paper is not honest to its readers. I understand the authors speak the lingo of integrable quantum systems but they are abusing it, making it seem like they have solved the relative quantum problem, which they have not.
As they say at the beginning of Section 2: "Extended quantum systems in very excited states are prohibitively challenging to be simulated for long times [11]..." So I think the paper deserves publication, and it is also reasonably well written, but the writing requires fine tuning to avoid making the reader wonder if the authors have solved a"prohibitively challenging" problem, which they have not.
Another point: while in the quantum transport set-up driven-dissipative boundaries (which lead to the Lindblad equation) are used to mimic the leads attached to a mesoscopic object (in the markovian limit etc.), in classical mechanics I am not sure this can be easily justified nor I have seen it used before (no reference is quoted here, so I assume this is an invention of the authors). The authors should justify this set-up more convincingly.

Requested changes

1- Make it clear this is a paper about a classical problem.
2- Make more connections with other classical systems works, going outside the realm of quantum integrable systems.
3- Justifications of the driven-dissipative setup.

  • validity: high
  • significance: high
  • originality: top
  • clarity: good
  • formatting: perfect
  • grammar: excellent

Author:  Alvise Bastianello  on 2021-12-01  [id 1996]

(in reply to Report 1 on 2021-10-27)

In her/his report, the referee raised some points which helped us in further clarifying our exposition: we are thankful for the useful input. Hereafter we address in detail the questions of the referee and the consequent improvements on the manuscript. We hope the revised version of our work can satisfy the Referee's request and meet the publication criteria of Scipost Physics.

Detailed comments:

1) Referee: "This is a nice and original work on a classical 1-d model written as if it were a paper on a quantum model, which makes reading it quite confusing. It considers a classical system of quadratic oscillators with a non-linear potential confined in a region of size L."

1) Answer: We apologize for the misunderstanding, of course it was not our intention to confuse the reader. As stated already in the previous version of the manuscript, we focus on the classical setup to have access to large-scale numerical simulations which are beyond the reach of quantum numerical methods. Nonetheless, the microscopic mechanism at the origin of the boundary generalized resistance is explained in terms of simple kinetic arguments which are quantitatively captured by the Boltzmann approach (which can be formulated within the quantum framework as well). Since we look at the classical model just as a convenient tool to check our general picture and the same mechanism is expected to hold in the quantum scenario, we did not sufficiently advertise the use of the classical setup. In the revised version of the manuscript, we repair this mistake through the following changes

1a) In the abstract, we replace "quasiparticles"-> "carriers". While we think the concept of quasiparticle can be used in the classical scenario as well, we agree the reader could be induced to believe we are focusing mostly on quantum models. For the same reason, in the abstract we now explicitly added "Focusing on classical systems, we study..."

1b) We explicitly state in the introduction that we focus on the classical scenario. Nevertheless, let us point us that classical physics has been an irreplaceable tool to assess the validity of non-equilibrium scenarios also in connection with integrability, as shown for example by Refs. [54-57] of the revised manuscript.

1c) We changed the title of Section 2 to make the use of a classical model explicit.

1d) In Eq.(1), $\psi^\dagger \to \psi^$. The use of $\psi^\dagger$ instead of $\psi^$ was a typo, indeed it never appeared again in the manuscript.

1e) Below Eq. 3, we removed the explicit mention to the Wigner distribution. Even though the local phase-space distribution of the classical model is defined similarly to the quantum Wigner distribution, we agree it could be confusing for the reader.

2) Referee: "Another point: while in the quantum transport set-up driven-dissipative boundaries (which lead to the Lindblad equation) are used to mimic the leads attached to a mesoscopic object (in the markovian limit etc.), in classical mechanics I am not sure this can be easily justified nor I have seen it used before (no reference is quoted here, so I assume this is an invention of the authors). The authors should justify this set-up more convincingly."

2) Answer: To the best of our knowledge, we are not aware of the use of this strategy within the classical scenario and we could not find a proper reference. This is also the main motivation behind appendix A, which provides a detailed account of how the strategy works. In the revised version of the manuscript, we summarize in a few words the general idea before digging into the details: essentially, outgoing carriers flowing out of the defect are removed from the system thanks to the dissipation. On the other hand, the tunable drive allows injecting an excitation distribution of our choice. The effect of the boundaries is then captured by Eq. 16: numerically solving for the stationary solution $\partial_t n=0$ allows for a determination of the injected mode density as a function of the tunable drive $s(k)$. This allows for an exact characterization of the injected mode density in terms of the boundary noise: for example, in Fig. 3 the injected mode density $(k>0)$ of the microscopic simulations (symbols) is exactly tuned to the target value (solid curve) obtained by solving Eq. 16.

---

## Round 3 · Author Response

Dear Editor,

we thank you for your time in considering our submitted manuscript "Transport through interacting defects and lack of thermalisation" for publication in Scipost Physics. In her/his report, the referee raised some points which helped us in further clarifying our exposition: we are thankful for the useful input. Hereafter we address in detail the questions of the referee and the consequent improvements on the manuscript. We hope the revised version of our work can satisfy the Referee's request and meet the publication criteria of Scipost Physics.

Detailed comments:

1) Referee: "This is a nice and original work on a classical 1-d model written as if it were a paper on a quantum model, which makes reading it quite confusing. It considers a classical system of quadratic oscillators with a non-linear potential confined in a region of size L."

1) Answer: We apologize for the misunderstanding, of course it was not our intention to confuse the reader. As stated already in the previous version of the manuscript, we focus on the classical setup to have access to large-scale numerical simulations which are beyond the reach of quantum numerical methods. Nonetheless, the microscopic mechanism at the origin of the boundary generalized resistance is explained in terms of simple kinetic arguments which are quantitatively captured by the Boltzmann approach (which can be formulated within the quantum framework as well). Since we look at the classical model just as a convenient tool to check our general picture and the same mechanism is expected to hold in the quantum scenario, we did not sufficiently advertise the use of the classical setup. In the revised version of the manuscript, we repair this mistake through the following changes

  • In the abstract, we replace "quasiparticles"-> "carriers". While we think the concept of quasiparticle can be used in the classical scenario as well, we agree the reader could be induced to believe we are focusing mostly on quantum models. For the same reason, in the abstract we now explicitly added "Focusing on classical systems, we study..."

-We explicitly state in the introduction that we focus on the classical scenario. Nevertheless, let us point us that classical physics has been an irreplaceable tool to assess the validity of non-equilibrium scenarios also in connection with integrability, as shown for example by Refs. [54-57] of the revised manuscript.

-We changed the title of Section 2 to make the use of a classical model explicit.

  • In Eq.(1), \psi^\dagger -> \psi^. The use of \psi^\dagger instead of \psi^ was a typo, indeed it never appeared again in the manuscript.

  • Below Eq. 3, we removed the explicit mention to the Wigner distribution. Even though the local phase-space distribution of the classical model is defined similarly to the quantum Wigner distribution, we agree it could be confusing for the reader.

2) Referee: "Another point: while in the quantum transport set-up driven-dissipative boundaries (which lead to the Lindblad equation) are used to mimic the leads attached to a mesoscopic object (in the markovian limit etc.), in classical mechanics I am not sure this can be easily justified nor I have seen it used before (no reference is quoted here, so I assume this is an invention of the authors). The authors should justify this set-up more convincingly."

2) Answer: To the best of our knowledge, we are not aware of the use of this strategy within the classical scenario and we could not find a proper reference. This is also the main motivation behind appendix A, which provides a detailed account of how the strategy works. In the revised version of the manuscript, we summarize in a few words the general idea before digging into the details: essentially, outgoing carriers flowing out of the defect are removed from the system thanks to the dissipation. On the other hand, the tunable drive allows injecting an excitation distribution of our choice. The effect of the boundaries is then captured by Eq. 16: numerically solving for the stationary solution \partial_t n=0 allows for a determination of the injected mode density as a function of the tunable drive s(k). This allows for an exact characterization of the injected mode density in terms of the boundary noise: for example, in Fig. 3 the injected mode density (k>0) of the microscopic simulations (symbols) is exactly tuned to the target value (solid curve) obtained by solving Eq. 16.

---

## Editorial Decision

published